# Surgical treatment in the chronic phase for uncomplicated Stanford type B aortic dissection

**Akihito Matsushita**[1,2]*, **Minoru Tabata**[3], **Takashi Hattori**[2], **Wahei Mihara**[2], **Yasunori Sato**[4]

**1** Department of Cardiovascular Surgery, Teikyo University Chiba Medical Center, Chiba, Japan, **2** Department of Cardiovascular Surgery, Seikeikai Chiba Medical Center, Chiba, Japan, **3** Department of Cardiovascular Surgery, Juntendo University Graduate School of Medicine, Tokyo, Japan, **4** Department of Biostatistics, Keio University School of Medicine, Tokyo, Japan

* matsushita.akihito.hi@med.teikyo-u.ac.jp

**Data Availability Statement:** All relevant data are within the paper and its Supporting Information files.

**Funding:** The authors received no specific funding for this work.

## Abstract

### Background

The most appropriate surgical method for patients with uncomplicated type B aortic dissection (UTBAD) in the chronic phase remains controversial. This study evaluated the outcomes of patients with UTBAD who needed aortic treatment as well as the impact of the treatment method or indication criteria on their prognosis.

### Methods

This retrospective review of 106 consecutive patients with aortic events in the chronic phase who underwent initial treatment for UTBAD between 2004 and 2021 comprised three groups: 19 patients who underwent endovascular repair (TEVAR), 38 who underwent open aortic repair and the medication group that included 49 patients. Aortic events were defined as a late operation or indication for operation for dissected aorta, aortic diameter (AD) $\geq$ 55 mm, rapid aortic enlargement ($\geq$5 mm/6 months), and saccular aneurysmal change. The endpoint was all-cause death. We assessed the association between treatment methods or surgical indication criteria and mortality using a Cox regression analysis.

### Results

The 5-year actuarial mortality rates were 27.1% in the TEVAR group, 19.6% in the open aortic repair group, and 38.4% in the medication group ($p = 0.86$). Moreover, the 5-year actuarial mortality rates in patients who had AD $\geq$ 55 mm were significantly higher than those patients with other surgical indication criteria (41.2% vs. 18.7%, p < 0.01). Multivariable analysis revealed a significant difference in AD $\geq$ 55 mm (hazard ratio [HR]: 2.88, 95% confidence interval [CI] 1.38–6.02, $p < 0.01$) and age (HR: 1.09, 95% CI 1.05–1.13, $p < 0.01$).

### Conclusions

Under the existing surgical indication criteria, there was no difference in mortality rates among patients with UTBAD based on their surgical treatment.

**Competing interests:** The authors have declared that no competing interests exist.

**Abbreviations:** AD, aortic diameter; CI, confidence interval; CT, computed tomography; HR, hazard ratio; OR, open aortic repair; TEVAR, thoracic endovascular aortic repair; ULP, ulcer-like projection; UTBAD, uncomplicated type B aortic dissection.

## Introduction

In patients with uncomplicated type B aortic dissection (UTBAD), the most appropriate surgical treatment method in the chronic phase remains controversial. For instance, a recent meta-analysis revealed that thoracic endovascular aortic repair (TEVAR) for chronic UTBAD is associated with significant early benefits; however, these beneficial effects were not sustained at midterm [1]. Although some studies have reported that reintervention is more frequent in TEVAR than in open aortic repair (OR) [2], OR is not exempt from reintervention or late rupture [3].

OR in the extended region is invasive in some patients, and TEVAR is complicated as false lumen flow has to be controlled owing to anatomical factors [4]. Herein, we evaluated the outcomes of patients with UTBAD who require aortic care in the chronic phase and attempted to evaluate the impact on survival prognosis based on the treatment method or surgical indication criteria.

## Materials and methods

### Ethics statement and study design

The study protocol was in accordance with the Declaration of Helsinki and based on the STROBE Statement. The target sample size in the study design was approximately 100 cases based on the similarity of previous studies [1, 2]. The study was approved by the Institutional Review Boards of Seikeikai Chiba Medical Center (approval no. CMC 2019–7, 2023–1), and an informed consent waiver was obtained.

In this study, 310 consecutive patients with acute type B aortic dissection were enrolled between October 2004 and August 2021 from two centers in Japan. The following cases were excluded: 11 cases with nonacute type B aortic dissection that were diagnosed ≥2 weeks after symptom onset, 52 complicated cases (involving rupture, impending rupture, and malperfusion), and 137 cases without aortic events during the chronic phase. Additionally, four cases were excluded owing to their critical condition at the time of aortic events. One died from rupture, one from intestinal bleeding, and two from pneumonia; hence, they were unable to undergo surgery.

In this study, we retrospectively reviewed 106 consecutive patients with UTBAD who experienced aortic events in the chronic phase. Aortic events were defined as a late operation or indication for operation for dissected aorta, aortic diameter (AD) ≥ 55 mm, rapid aortic enlargement (≥5 mm/6 months), and saccular aneurysmal change. Patients were categorized into three groups for this study: the TEVAR group consisting 19 patients who underwent TEVAR, the OR group comprising 38 patients who underwent OR, and the medication group comprising 49 patients who received medical therapy alone (Fig 1). The primary reasons for patients being in the medication group were their refusal to undergo invasive treatment or being considered inoperable owing to old age or frailty.

The study aimed to assess the all-cause mortality as the endpoint and compare the outcomes between the three groups. Furthermore, the study analyzed and compared the outcomes for each pair of treatment (TEVAR vs. medication, TEVAR vs. OR, and medication vs. OR) using propensity score matching. Moreover, it investigated whether the outcome was influenced by the time periods or surgical indication criteria.

Final follow-up data were collected between November 2021 and June 2022 via the medical computer records system survey of two centers. Follow-up data were also collected for outpatients in another hospital via telephone and mail interviews. In addition, outpatient visits that were lost to follow-up at their last appointment were surveyed. During data collection,

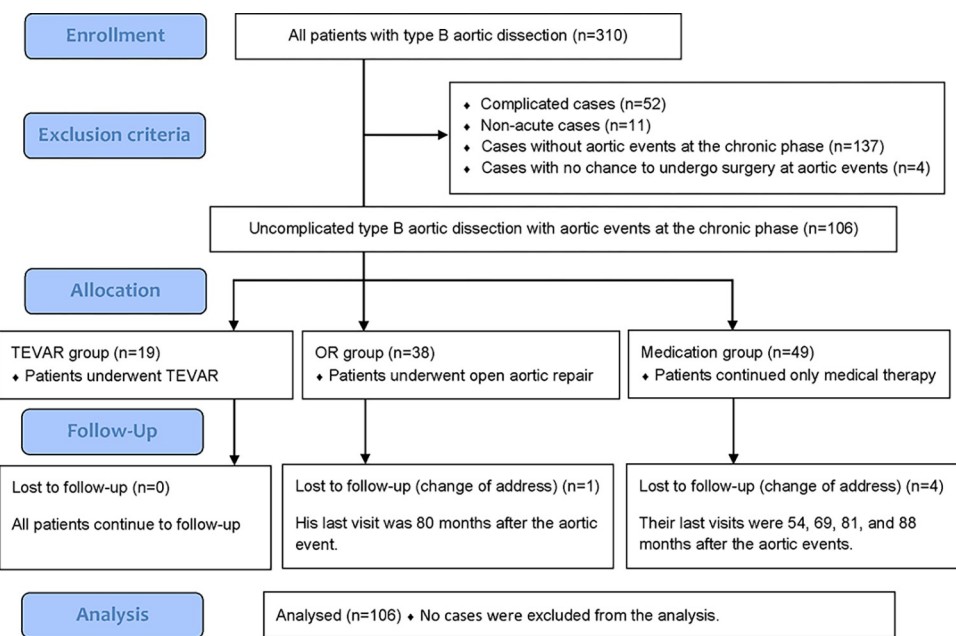

**Fig 1. Patient selection flow diagram.** Flow diagram of the entire series of patients with Stanford type B aortic dissection. TEVAR, thoracic endovascular aortic repair.

information that could identify individual participants was accessible. However, after this stage, personal information was separated from the data to ensure privacy and anonymity. These data were finally accessed for research purpose in April 2023.

## Initial therapy protocol for UTBAD

All patients were diagnosed using multiphasic computed tomography (CT). Per the Stanford classification, aortic dissection was classified as type B it did not involve the ascending aorta. Herein, the study group included type B intramural hematoma because it was difficult to distinguish intramural hematoma from aortic dissection with a totally thrombosed false lumen. Our acute aortic dissection protocol has been described previously [5, 6]. All patients with UTBAD were intended to be managed medically, regardless of their false lumen status or the AD. Contrast CT scans were performed during follow-up visits after 3–6 months, 1 year, and annually thereafter. After discharge, the patients were continued to be followed up to ensure a systolic blood pressure of <120 mmHg.

## Protocol for late aortic events

In our institution, TEVAR was introduced for dissected thoracic aneurysm and J Graft Open Stent Graft℗ (Japan Lifeline, Tokyo, Japan) for frozen elephant trunk graft in 2016. The time period under consideration was categorized as follows: 2004–2015 (the early period before the introduction of the stent graft) and 2016–2021 (the late period after the stent graft). As expected, TEVAR was adopted more frequently in the late period. If surgery was not selected in either time period, the patient continued with medication. In each time period, patient enrollment in the OR, TEVAR, and medication groups occurred concurrently.

In the early period, OR was regarded as the primary treatment approach in our institution because it directly addressed the dissected aortic aneurysm. However, our surgical plan was modified in the late period in response to changes in guidelines [7]. We selected TEVAR for

patients who had suitable anatomy, healthy and adequate distance of the landing zone, suitable landing size, and healthy access. OR was the secondary treatment approach in the late period. For patients with ascending aorta aneurysm or anatomy unsuitable for TEVAR, we performed total aortic arch replacement using the frozen elephant trunk procedure. Additionally, we performed descending aorta or thoracoabdominal replacement depending on the aneurysm position. Every patient in the treatment group underwent either planned or elective surgery.

## Data collection and definitions

We collected in-hospital data from the patients' medical records. UTBAD was defined as follows according to previous European Society of Cardiology guidelines [7]: unruptured dissected aorta, no impending rupture of the dissected aorta (continuous symptoms despite optimal medical treatment with antipain or anti-impulse medication), and absence of malperfusion (any newly developed symptoms with the presence of false lumen expansion and impaired true lumen flow on the CT images). Moreover, stroke was characterized as a lasting central neurologic deficit persisting for >72 hours after the surgery. All strokes were verified with CT scans. Acute kidney injury was defined as an increase in serum creatinine of $\geq 0.3$ mg/dL or $\geq 50\%$ within 1 week after the surgery. In-hospital mortality was defined as death before discharge. Aortic-cause mortality included all deaths related to the treated pathology, including aneurysm rupture, aortic dissection retrograde progression, operative procedure complications, or clinical suspicion of aortic death without another leading cause.

AD and false lumen thickness and patency were measured in initial CT images. The largest diameters of the long and short axes and false lumen thickness were measured at five sites according to the zone system, as previously mentioned [5, 6].

Patent false lumen was defined as any contrast effect in the false lumen in the early or late vascular phase, except for ulcer-like projection (ULP). ULP was defined as focal, well-defined pouches of contrast medium measuring $\leq 10$ mm in length and projecting into the noncommunicating false lumen along the long axis of the aorta. Additionally, ULP enlargement or aneurysmal formation was regarded as a type of saccular aneurysm.

## Statistical analysis

For patient characteristics and CT data, summary statistics were constructed using frequencies and proportions for categorical data and means and standard deviations or median and interquartile range, as appropriate, for continuous variables. Univariable analyses were performed using the Fisher's exact test or Wilcoxon test for continuous variables. All potential predictors were entered into the univariate analyses. To detect the mortality risk factors, variables with a P value of <0.25 and major variables from the previous literature were then selected for the propensity score match and the multivariate model.

The following variables were incorporated into the propensity score calculations: age, male sex, peripheral artery disease, chronic obstructive pulmonary disease, treatment center, time from onset of dissection to aortic events, initial $AD \geq 40$ mm, ULP, $AD \geq 55$ mm, and saccular aneurysmal change. Matching was performed using a greedy 5-to-1 digit-matching algorithm. Following propensity score matching, 17 matched pairs (TEVAR vs. medication), 13 matched pairs (TEVAR vs. OR), and 21 matched pairs (medication vs. OR) were established.

For time-to-event outcomes, the lengths of time from the onset of aortic events to mortality were compared using a log-rank test, whereas the Kaplan–Meier method was used to estimate the absolute risk of mortality. Hazard ratios (HRs) and 95% confidence intervals (CIs) were estimated using the Cox proportional hazards model. To identify baseline and clinical variables associated with overall survival time after aortic events, a multivariable analysis was

performed using the Cox proportional hazard model, considering several factors as covariates. These factors included age, male sex, peripheral artery disease, chronic obstructive pulmonary disease, treatment center, AD $\geq$ 55 mm, rapid aortic expansion, and treatment methods.

All comparisons were planned, and the tests were two-sided. A $p$ value of $<0.05$ was considered statistically significant. All statistical analyses were initially performed using JMP, version 14.2.0 (SAS Institute Inc., Cary, NC); SAS software program, version 9.4 (SAS Institute Inc.); SPSS, version 22.0 (IBM-Corp., Armonk, NY); and R, version 3.00.

## Results

### Patient characteristics

Table 1 presents the patient characteristics and includes all data. The physician preferred TEVAR over OR at the first center. However, there was no significant difference in other factors.

Table 2 summarizes the CT measurement data at onset. Both aortic minor and major axis diameters at the largest site were larger in the OR group than in the other groups. Patients with an initial AD $\geq$ 40 mm were more frequent in the OR group than in the other groups. The indication for surgery was as mentioned above and summarized in Table 2.

### TEVAR

The purpose of TEVAR was the primary entry closure in 10 patients with a patent false lumen. We also attempted to close the reentry if possible. We performed false lumen flow control using a vascular plug in the false lumen unless the reentry was closed because it presented under the celiac artery level. Furthermore, in nine patients with thrombosed false lumen, we also performed TEVAR to treat dissected aortic aneurysm. The primary proximal landing zone was as follows: nondissected zone 3 for 14 patients, zone 2 for 1 patient who underwent a right axillary artery–left axillary artery bypass, zone 2 for 3 patients who had simple left subclavian artery coil embolization, and zone 0 for 1 patient who underwent total debranch bypass

**Table 1. Characteristics of the patients.**

| Variables | TEVAR group (n = 19) | OR group (n = 38) | Medication group (n = 49) | P-value TEVAR vs. OR | P-value TEVAR vs. Medication | P-value OR vs. Medication |
|---|---|---|---|---|---|---|
| Age at aortic dissection onset (years; mean ±SD) | 64.9±14.3 | 66.3±9.4 | 66.1±14.3 | 0.714 | 0.742 | 0.948 |
| Age at aortic event (years; mean ±SD) | 66.4±13.9 | 67.4±9.3 | 67.4±14.7 | 0.760 | 0.770 | 0.975 |
| Male | 15 (78.9%) | 30 (78.9%) | 38 (77.6%) | 1.000 | 1.000 | 1.000 |
| Hypertension | 18 (94.7%) | 38 (100%) | 49 (100%) | 0.333 | 0.279 | 1.000 |
| Hyperlipidemia | 3 (15.8%) | 10 (26.3%) | 10 (20.4%) | 0.510 | 1.000 | 0.610 |
| Diabetes mellitus | 1 (5.3%) | 5 (13.2%) | 5 (10.2%) | 0.652 | 1.000 | 0.742 |
| Peripheral artery disease | 2 (10.5%) | 5 (13.2%) | 1 (2.0%) | 1.000 | 0.187 | 0.082 |
| Cerebral infarction | 2 (10.5%) | 3 (7.9%) | 4 (8.2%) | 1.000 | 0.669 | 1.000 |
| Chronic obstructive pulmonary disease | 2 (10.5%) | 3 (7.9%) | 0 (0%) | 1.000 | 0.075 | 0.080 |
| De Bakey classification type IIIa | 8 (42.1%) | 15 (39.5%) | 16 (32.7%) | 1.000 | 0.574 | 0.652 |
| Treatment at 1st Center | 15 (78.9%) | 13 (34.2%) | 29 (59.2%) | 0.002 | 0.163 | 0.030 |
| Time from dissection onset to aortic events (months) | 6 [2–18] | 7 [3–14] | 5 [3–12] | 0.872 | 0.800 | 0.803 |

SD: standard deviation; DeBakeyIIIa: Aortic dissection stops above the diaphragm

**Table 2. CT data at aortic dissection onset and aortic event criteria.**

| Variables | TEVAR cases (n = 19) | OR cases (n = 38) | Medication group (n = 49) | P-value TEVAR vs. OR | P-value TEVAR vs. Medication | P-value OR vs. Medication |
|---|---|---|---|---|---|---|
| Aortic minor axis diameter at largest site (mm; mean ±SD) | 38.8±8.2 | 44.1±9.0 | 38.4±7.8 | 0.025 | 0.839 | 0.002 |
| Aortic major axis diameter at largest site (mm; mean ±SD) | 41.5±8.9 | 46.1±9.5 | 40.2±8.1 | 0.064 | 0.566 | 0.002 |
| False lumen diameter at largest site (mm; mean ±SD) | 12.7±5.7 | 13.9±8.6 | 12.4±6.4 | 0.550 | 0.862 | 0.321 |
| Aortic minor axis diameter ≥ 40mm | 9 (47.4%) | 30 (78.9%) | 24 (49.0%) | 0.032 | 1.000 | 0.007 |
| Thrombosed false lumen | 9 (47.4%) | 20 (52.6%) | 21 (42.9%) | 0.783 | 0.790 | 0.394 |
| Ulcer like projection (1 week after onset) | 7 (36.8%) | 5 (13.2%) | 15 (30.6%) | 0.081 | 0.774 | 0.073 |
| Surgical indication criteria as aortic events | | | | | | |
| Aortic diameter ≥ 55 mm | 8 (42.1%) | 22 (57.9%) | 13 (26.5%) | 0.279 | 0.249 | 0.004 |
| Rapid aortic enlargement (≥ 5 mm/6 months) | 5 (26.3%) | 14 (36.8%) | 19 (38.8%) | 0.555 | 0.405 | 1.000 |
| Saccular aneurysmal change | 6 (31.6%) | 2 (5.3%) | 17 (34.7%) | 0.013 | 1.000 | 0.001 |

SD: standard deviation

involving the ascending aorta to the brachiocephalic artery, left common carotid artery, and left subclavian artery using a three-branched Hemashield® vascular graft. Depending on the aortic status, access rout diameter, and landing distance, we chose from six devices (Conformable GORE® TAG® for three patients, Relay PLUS® for two patients, VALIANT CAPTI-VIA® for eight patients, Cook Zenith Dissection® for three patients, Zenith TX 2® for two patients, and Zenith alpha® for one patient). Particularly, the Zenith Dissection endovascular stent device was implanted in five patients with rapid aortic expansion in the subacute phase. The mean device size was 36.2 ± 5.9 mm for the proximal site and 31.7 ± 5.6 mm for the distal site, the median operation time was 104 min (range: 82–134 min), and the radiation fluoroscopy time was 17 min (range: 14–26 min).

## OR

The range of replacement due to the dissected AD was determined. Our main aim in performing open surgery was to replace the primary entry tear and dissected aorta with a diameter of >4.0 cm. We performed less extensive replacement using double-barrel distal anastomosis for all patients with a patent false lumen. A total of 16 patients with a double-barrel distal anastomosis site in the dissected aorta with a diameter of <4.0 cm underwent descending dissected aorta replacement. In addition, simple total aortic arch replacement was performed in 9 patients, whereas staged operations were considered in 11 patients with concomitant arch pathology. In all these 11 patients, total aortic arch replacement with frozen elephant trunk (J Graft Open Stent Graft®) was performed as the first surgery, Furthermore, of these 11 patients, 3 underwent descending aorta replacement and 3 underwent TEVAR as the second surgery. The remaining five patients were monitored without second surgery. Two patients underwent abdominal dissected aorta replacement with double-barrel proximal anastomosis.

## Medical therapy

Six of our patients were deemed inoperable because of their high-risk status: four were octogenarians, one was a nonagenarian, and one had cancer. Three of the patients had AD ≥ 55 mm.

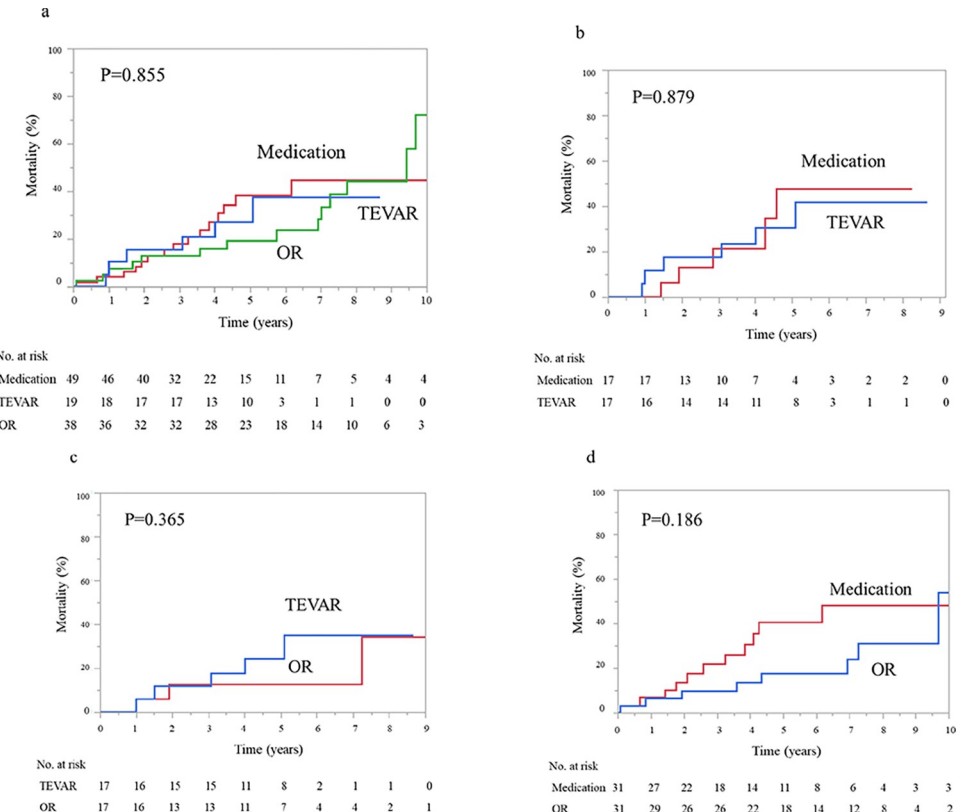

**Fig 2. Kaplan–Meier curve.** (a) All-cause mortality for patients in the TEVAR, OR, and medication groups. (b) All-cause mortality for propensity-matched patients who underwent TEVAR and received medication. (c) All-cause mortality for propensity-matched patients who underwent TEVAR and OR. (d) All-cause mortality for propensity-matched patients who received medication and underwent OR. TEVAR, thoracic endovascular aortic repair; OR, open aortic repair.

We strongly recommended surgical intervention for ten patients with AD $\geq$ 55 mm, nine patients with rapid aortic enlargement, and six patients with saccular aneurysms. Nonetheless, despite our recommendations, all these patients refused surgery and opted to continue with medication-only treatment. In contrast, the decision to continue medication was left to the discretion of the outpatient doctors for eight patients with rapid aortic enlargement and ten patients with saccular formation of ULP.

## Short-term outcomes

In-hospital mortality owing to intestinal bleeding occurred in only one patient (2.9%) who underwent OR. Postoperative morbidities in patients who underwent TEVAR included one case of paraparesis, one case of stroke, and one case of acute kidney injury. In contrast, patients who underwent OR included eight cases of acute kidney injury, two cases of paraparesis, one case of intestinal bleeding, one case of mediastinitis, and one case of chylothorax. Notably, none of the patients who developed acute kidney injury required hemodialysis.

## Long-term outcomes

The median duration of follow-up after aortic events was 53 months (interquartile range 31–74 months), and the follow-up rate was 95.3%. Five patients were lost to follow-up because of

**Table 3. Characteristics of the patients after propensity score matching.**

| Variables | TEVAR group (n = 17) | Medication group (n = 17) | P-value TEVAR vs. Medication | TEVAR group (n = 13) | OR group (n = 13) | P-value TEVAR vs. OR | Medication group (n = 21) | OR group (n = 21) | P-value Medication vs. OR |
|---|---|---|---|---|---|---|---|---|---|
| Age at aortic dissection onset (years; mean ± SD) | 65.4 ± 14.0 | 68.4 ± 13.5 | 0.521 | 64.2 ± 14.2 | 67.0 ± 7.8 | 0.574 | 67.6 ± 14.0 | 63.7 ± 9.7 | 0.305 |
| Age at aortic event (years; mean ± SD) | 66.9 ± 13.3 | 70.2 ± 14.3 | 0.492 | 66.0 ± 15.7 | 68.7 ± 8.2 | 0.588 | 69.1 ± 14.6 | 65.3 ± 9.8 | 0.332 |
| Male | 13 (76.5%) | 12 (70.6%) | 1.000 | 9 (69.2%) | 9 (69.2%) | 1.000 | 16 (76.2%) | 17 (81.0%) | 1.000 |
| Hypertension | 16 (94.1%) | 17 (100%) | 1.000 | 12 (92.3%) | 13 (100%) | 1.000 | 21 (100%) | 21 (100%) | 1.000 |
| Hyperlipidemia | 3 (17.7%) | 1 (5.9%) | 0.601 | 3 (23.1%) | 3 (23.1%) | 1.000 | 6 (28.6%) | 3 (14.3%) | 0.454 |
| Diabetes mellitus | 1 (5.9%) | 1 (5.9%) | 1.000 | 1 (7.7%) | 3 (23.1%) | 0.593 | 1 (4.8%) | 3 (14.3%) | 0.606 |
| Peripheral artery disease | 1 (5.9%) | 1 (5.9%) | 1.000 | 2 (15.4%) | 3 (23.1%) | 1.000 | 1 (4.8%) | 2 (9.5%) | 1.000 |
| Cerebral infarction | 2 (11.8%) | 1 (5.9%) | 1.000 | 1 (7.7%) | 2 (15.4%) | 1.000 | 2 (9.5%) | 2 (9.5%) | 1.000 |
| Chronic obstructive pulmonary disease | 0 (0%) | 0 (0%) | 1.000 | 2 (15.4%) | 1 (7.7%) | 1.000 | 0 (0%) | 0 (0%) | 1.000 |
| De Bakey classification type IIIa | 8 (47.1%) | 5 (29.4%) | 0.481 | 4 (30.8%) | 5 (38.5%) | 1.000 | 9 (42.9%) | 7 (33.3%) | 0.751 |
| Treatment at 1st Center | 13 (76.5%) | 13 (76.5%) | 1.000 | 9 (69.2%) | 10 (76.9%) | 1.000 | 11 (52.4%) | 11 (52.4%) | 1.000 |
| Time from dissection onset to aortic events (months) | 6 [2–22] | 5 [3–36] | 0.972 | 7 [2–36] | 7 [4–17] | 0.878 | 5 [3–13] | 7 [3–17] | 0.850 |

SD: standard deviation; DeBakeyIIIa: Aortic dissection stops above the diaphragm

change of address. Additional surgeries or multiple surgeries were required in four patients (21.1%) in the TEVAR group and eight patients (21.1%) in the OR group. The incidence of all-cause mortality was 6 patients (31.6%) in the TEVAR group, 14 patients (36.8%) in the OR group, and 15 patients (30.6%) in the medication group. Of these patients, aortic-cause mortality was observed in three patients (50.0%) in the TEVAR group (type A aortic dissection in two and operative procedure complications in one), and eight patients (57.1%) in the OR group (rupture in three, type A aortic dissection in one, operative procedure complications in two, and sudden death in two). In the medication group, aortic-cause mortality occurred in seven patients (46.7%) (rupture in five and sudden death in two). There were no significant differences in all-cause mortality or aortic-cause mortality among these groups.

The proportions of actuarial mortality at 1, 3, and 5 years were 10.5%, 15.8%, and 27.1% in the TEVAR group; 7.9%, 13.3%, and 19.6% in the OR group; and 4.2%, 18.2%, and 38.4% in the medication group ($p$ = 0.855; Fig 2A), respectively.

## Outcomes of propensity score-matched patients

Table 3 lists the characteristics of the propensity-matched patients. These matched pairs were well balanced for all covariates, with no significant differences in characteristics. Actuarial mortality at 1, 3, and 5 years occurred in 11.8%, 17.7%, and 30.5% in the TEVAR group and in 6.3%, 25.0%, and 35.7% in the medication group ($p$ = 0.912; Fig 2B), respectively. In addition, no significant differences were found in actuarial mortality between the TEVAR and OR groups ($p$ = 0.717; Fig 2C), or between the medication and OR groups ($p$ = 0.133; Fig 2D) after propensity score matching.

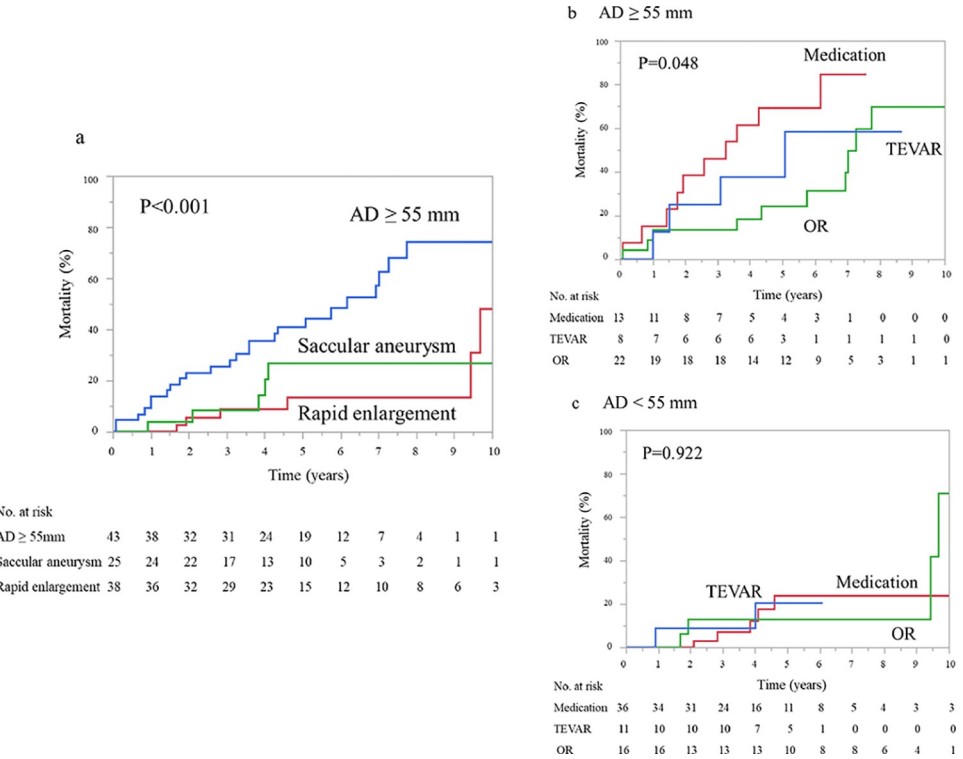

**Fig 3. Kaplan–Meier curve based on surgical indication criteria.** (a) All-cause mortality among patients with AD ≥ 55 mm, patients with saccular aneurysmal change, and patients with rapid aortic enlargement. (b) All-cause mortality of patients with AD ≥ 55 mm in the TEVAR, OR, and medication groups. (c) All-cause mortality of patients with AD < 55 mm in the TEVAR, OR, and medication groups. AD, aortic diameter; TEVAR, thoracic endovascular aortic repair; OR, open aortic repair.

## Outcomes based on surgical indication criteria

Moreover, we compared the mortality according to surgical indication. From the point of view of surgical indication criteria, the mortality rate of patients with AD ≥ 55 mm (55.8%, four in the TEVAR group, ten in the OR group, and ten in the medication group) was higher than in patients with rapid aortic enlargement (15.8%, four in the OR group and two in the medication group) and in patients with saccular aneurysm (20.0%, two in the TEVAR group and three in the medication group).

The proportions of actuarial mortality at 1, 3, and 5 years were 13.9%, 25.7%, and 41.2% in patients with AD ≥ 55 mm; 4.0%, 8.4%, and 26.7% in those with saccular aneurysmal change; and 0%, 8.8%, and 13.6% in those with rapid aortic enlargement ($p < 0.001$; Fig 3A), respectively. In addition, the proportions of actuarial mortality at 1, 3, and 5 years were 13.6%, 13.6%, and 24.5% in the OR group; 12.5%, 25.0%, and 37.5% in the TEVAR group; and 15.4%, 46.2%, and 69.2% in the medication group for patients with AD ≥ 55 mm ($p = 0.048$; Fig 3B). However, there was no significant difference among the three groups of AD < 55 mm with rapid aortic enlargement or saccular aneurysmal change($p = 0.922$; Fig 3C).

## Outcomes depending on time periods

Furthermore, we compared mortality based on different time periods. The median duration of follow-up after aortic events was 83 months (interquartile range 42–103 months) in the early period (2004–2015) and 46 months (interquartile range 26–58 months) in the late period

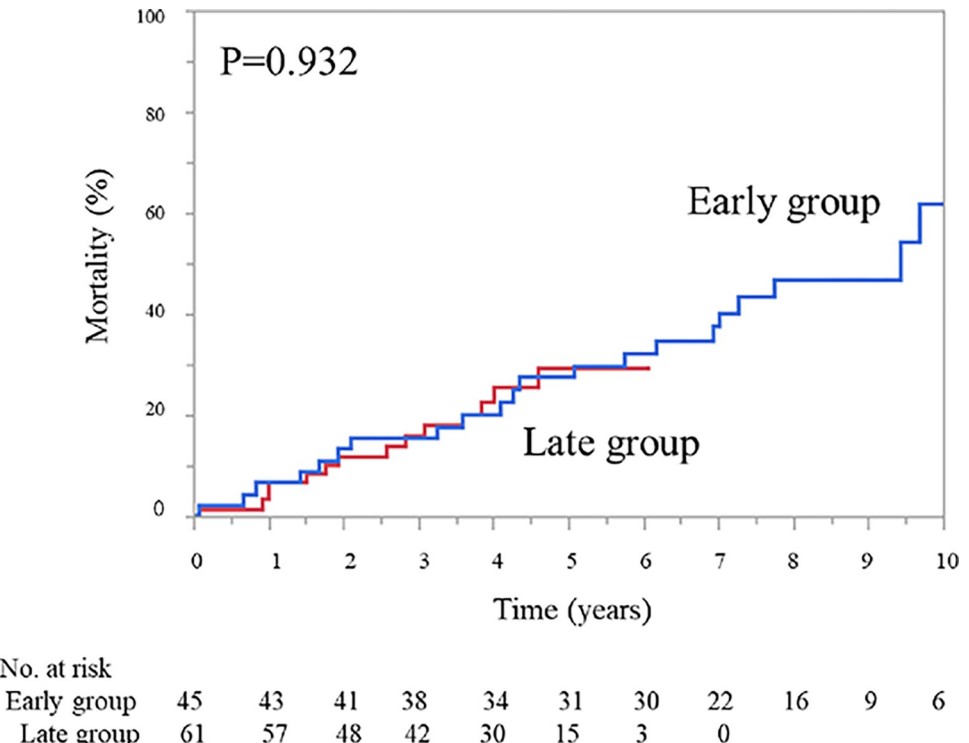

**Fig 4. Kaplan–Meier curve based on different time periods.** All-cause mortality for patients in the early (2004–2015) and late periods (2016–2021).

(2016–2021) ($p < 0.001$). The results showed no significant differences between any of the patients in the early (2004–2015) and late (2016–2021) periods ($p = 0.932$; Fig 4).

## Multivariable analysis

Multivariable Cox analysis revealed no significant difference in the therapeutic effects of TEVAR (HR = 1.329, 95% CI 0.448–3.938, $p = 0.608$) and OR (HR = 0.592, 95% CI 0.248–1.413, $p = 0.237$) but indicated a significant difference in AD $\geq$ 55 mm as surgical indication (adjusted HR = 2.876, 95% CI 1.375–6.016, $p = 0.005$) and age (adjusted HR = 1.087, 95% CI 1.047–1.128, $p < 0.001$; Table 4).

**Table 4. Multivariable analysis of factors associated with all-cause mortality.**

| Variables | Maximum model | | Final model | |
|---|---|---|---|---|
| | Hazard ratio (95% CI) | p value | Adjusted hazard ratio (95% CI) | p value |
| Age | 1.081 (1.041–1.122) | <0.001 | 1.087 (1.047–1.128) | <0.001 |
| Male sex | 0.759 (0.284–2.034) | 0.584 | | |
| Peripheral artery disease | 1.201 (0.351–4.104) | 0.770 | | |
| Chronic obstructive pulmonary disease | 0.788 (0.131–4.737) | 0.794 | | |
| Treatment at 1ˢᵗ Center | 0.639 (0.269–1.515) | 0.309 | | |
| Open aortic repair | 0.592 (0.248–1.413) | 0.237 | | |
| Thoracic endovascular repair | 1.329 (0.448–3.938) | 0.608 | | |
| Aortic diameter ≥55mm | 2.516 (0.884–7.157) | 0.084 | 2.876 (1.375–6.016) | 0.005 |
| Rapid aortic expansion (≥5 mm/6 months) | 0.706 (0.189–2.635) | 0.604 | | |

CI: confidence interval

## Discussion

In patients with UTBAD in the chronic phase, the treatment method remains controversial. Guidelines on therapeutic indications and methods, such as those indicating that TEVAR is the preferable treatment, have changed recently. Based on these changes, we changed our surgical plan in 2016. We selected TEVAR in patients with suitable anatomy. OR was considered as the secondary therapy. Before 2016, we diligently recommended surgical treatment primarily for patients with AD $\geq$ 55 mm. For patients with only initial rapid aortic enlargement or ULP saccular formation, we considered the operative risk to be higher than the rupture risk. Therefore, there were significant differences in surgical indication criteria between the treatment and medication groups in this study. Additionally, Roselli reported that many physicians and surgeons hesitate to recommend open surgery until it is deemed absolutely necessary, and patients fear open distal aortic repair. For these reasons, open surgery was often postponed until patients developed very late complications [8]. We did not observe any significant prognostic improvement based on different time periods although there was a serious time trend bias. However, it is worth noting that changes in our surgical plan and potential patient selection bias might have influenced the study outcomes.

We found several mortality cases in the medication group after the aortic event. Our results showed distinctly worse outcomes in patients with AD $\geq$ 55 mm who did not follow the surgical recommendation. It is difficult to expect a long-term prognosis for surgically indicated cases with only medication. However, we also found that almost half of the patients with AD $\geq$ 55 mm died within 7 years postoperatively, regardless of whether they underwent TEVAR or OR. In this study, patients with AD $\geq$ 55 mm had poor outcomes. Pujara et al. also reported greater maximum aortic diameter predicted poorer outcome in OR for chronic aortic dissected aneurysm [9]. Furthermore, we also found 11 cases (19.3%) of long-term aortic-cause mortality, even among patients who received TEVAR or OR. Additionally, 12 patients (21.1%) required a second intervention after initial TEVAR or OR. Therefore, we could not identify prognosis improvement with surgical treatment for aortic events in patients with UTBAD. Lou et al. reported no significant difference in long-term mortality among patients who received TEVAR in the chronic phase, open surgery, or optimal medication [10]. They further reported that TEVAR in the acute phase might confer a survival advantage [10]. Nevertheless, a recent meta-analysis and a large cohort study comparing endovascular and medical management for UTBAD concluded that the benefits of TEVAR in managing acute/subacute UTBAD remain uncertain [11, 12].

We previously documented the rapid dilatation of the dissected aorta within 6 months from dissection onset and its rate decrease gradually in patients with a false lumen larger than the true lumen [6]. In the present study, patients who continued only medication after rapid dilatation within the first 6 months from onset seldom experienced continuous rapid aortic growth, and there was no aortic-cause death in patients who had only rapid dilatation but without AD $\geq$ 55 mm or ULP. These patients refused both TEVAR and OR and had a good clinical course in our study period. However, it is possible to achieve good remodeling with TEVAR in the early phase in these cases. Controversy remains regarding how to treat such cases. The current indications and methods for surgical treatment are not the best options for every patient. Each patient's status and background should be considered when determining the most appropriate treatment method. Preemptive therapy may be considered for an immature dissected aortic aneurysm in young patients in whom the aortic aneurysm is expected to expand.

### Limitations

This study has several limitations. First, it was a retrospective observational study performed at two centers in Japan, and the sample size and variables were limited. There was a patient

selection bias, as mentioned. To address potential biases, we conducted multivariable analysis. In addition, via propensity score matching, we performed sensitivity analyses to compare each treatment. As mentioned earlier, these analyses revealed no significant differences between the therapeutic effects of TEVAR and OR. Second, we included UTBAD with both a thrombosed false lumen and type B intramural hematoma as these conditions are difficult to distinguish. Finally, we lost five patients (4.7%) during follow-up. However, this was not a selective drop-out, and the sample size was maintained at the previous study level. Our study is exploratory, which makes it difficult to calculate the detection power or to specify the desired precision for the crude estimates in advance. Nevertheless, as mentioned earlier, we maintained a sample size that was consistent with that of previous similar studies [1.2]. To clarify the long-term outcomes of surgical treatment in patients with UTBAD, a large prospective and external validation studies with better follow-up rates are needed.

## Conclusions

We could not identify an improvement in prognosis via surgical treatment for aortic events in patients with UTBAD under the current surgical indication criteria. Patients with AD $\geq$ 55 mm or advanced age had worse outcomes than those with AD < 55 mm.

These findings highlight the importance of not delaying treatment until the patients develop advanced diseases. Instead, surgical treatment methods should be thoughtfully and individually considered based on the specific status of each patient.

## Supporting information

**S1 Checklist.**
(DOCX)

**S1 Data.**
(XLSX)

**S2 Data.**
(DOCX)

**S3 Data.**
(DOCX)

**S1 File.**
(PDF)

## Acknowledgments

The authors would like to thank Enago (www.enagp.jp) for the English language review.

## Author Contributions

**Conceptualization:** Akihito Matsushita, Minoru Tabata, Yasunori Sato.

**Data curation:** Akihito Matsushita, Takashi Hattori, Wahei Mihara.

**Formal analysis:** Akihito Matsushita, Minoru Tabata, Yasunori Sato.

**Investigation:** Akihito Matsushita, Takashi Hattori, Wahei Mihara.

**Methodology:** Akihito Matsushita, Takashi Hattori, Wahei Mihara, Yasunori Sato.

**Project administration:** Wahei Mihara, Yasunori Sato.

**Software:** Akihito Matsushita, Yasunori Sato.

**Supervision:** Minoru Tabata, Takashi Hattori, Wahei Mihara, Yasunori Sato.

**Validation:** Akihito Matsushita, Minoru Tabata, Yasunori Sato.

**Visualization:** Akihito Matsushita.

**Writing – original draft:** Akihito Matsushita, Minoru Tabata, Takashi Hattori, Wahei Mihara, Yasunori Sato.

**Writing – review & editing:** Akihito Matsushita, Yasunori Sato.

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
