## [Decision Letter · Decision Letter 0]

14 Nov 2023

PONE-D-23-24756Surgical treatment at the chronic phase for uncomplicated Stanford type B aortic dissectionPLOS ONE

Dear Dr. Matsushita,

Thank you for submitting your manuscript to PLOS ONE. After careful consideration, we feel that it has merit but does not fully meet PLOS ONE’s publication criteria as it currently stands. Therefore, we invite you to submit a revised version of the manuscript that addresses the points raised during the review process.

We look forward to receiving your revised manuscript.

Kind regards,

Alessandro Leone, MD

Academic Editor

PLOS ONE

Reviewers' comments:

Reviewer's Responses to Questions

**Comments to the Author**

1. Is the manuscript technically sound, and do the data support the conclusions?

Reviewer #1: Yes

Reviewer #2: Yes

Reviewer #3: No

2. Has the statistical analysis been performed appropriately and rigorously? 

Reviewer #1: No

Reviewer #2: Yes

Reviewer #3: No

3. Have the authors made all data underlying the findings in their manuscript fully available?

Reviewer #1: No

Reviewer #2: Yes

Reviewer #3: No

4. Is the manuscript presented in an intelligible fashion and written in standard English?

Reviewer #1: Yes

Reviewer #2: No

Reviewer #3: No

5. Review Comments to the Author

Reviewer #1: Congratulations to authors for extremely interesting topic choice and writing accuracy.

In order to get the most out of scientific questions one should design a prospective randomized trial, but in this case it is extremely complicated due to the type of disease.

Materials

Why do you consider the TEVAR as a surgical group?

Results:

It seems that the surgical group has higher comorbidity?

Had bigger aorta

Had high risk status

Medical Group had higher mortality in the follow up?

Topic is super specialistic. I think you should compare the TEVAR with medical therapy alone, because it seems that some of the patients in the medical group could have been addressed to the TEVAR. Try to add propensity score matching to two groups of interest and do subanalysis.

Reviewer #2: The authors studied an important question on patients with progressive dilatation of Type B dissection or have reached an aortic size over 5.5 cm with our without any symptoms. They noticed that in patients with aortic diameter of over 5.5 cm there is benefit in intervention, rest of them did not benefit from intervention.

The paper is well written and made a sincere effort to address the question given the long duration of study and heterogeneity of population as well as use of TEVAR after 2016.

Few suggestions to authors to make the paper more easy for readers to follow.

1. Create a table that shows inclusion and exclusion criteria, make it simple

2. limit abbreviations to minimum- as a rule if you are not using the acronym for more than five times, spell it out. It is easier to stay focused on the message.

3. while the syntax and grammar are OK the flow of information is not. Please use a service to make the whole paper revised to improve readability.

I thank the authors for undertaking this important study.

Reviewer #3: The authors present the results of a cohort study of 106 consecutive patients consisting of 57 surgical patients and 49 medicational treated patients with UTBAD with respect to all cause mortality. They concluded, that there was difference in mortality between the surgical and medicational treated patients.

General comment:

The hypothesis, primary endpoint variables, statistical analysis and sample size justification are not aligned.

• The statement in Line 108implies, that the most severe case (inoperable) and the less severe cases (refusal for invasive treatment) were pooled in the medicational treated group. This heterogeneity may cause bias and may be explored by Kaplan Meier Curves in subgroups.

• Although this is an exploratory study, sample size justification in terms of stating the desired precision of the (crude) estimates should be given,

• The major problem of the study design is the missing randomization. This result in an unfair comparison of the treatment effects. The authors used multivariate models but the elaboration of confounders is incomplete. So uncertainty of the results might need further exploration.

• The analysis of the surgical group splits in TEVAR and OR technique. If this is of primary interest, the treatment factor should be split in 3 groups, medicational, TEVAR, OR. Then the statistical analysis of the time to event data should be made accordingly. Even here, without randomization a sound multivariate model building should be applied.

• Concerning the multivariate model building, one first should identify promising confounders, e.g Cox models for each confounder variable (see table 1 and 2). If p≤.25 one look to potential interactions. Of primary interest are the confounder times treatment group interactions as Cox regression. From this it is clear whether to include some interaction terms. The remaining set of all promising confounders and interaction terms give the whole Cox regression model. The interaction part is missing.

• Some results in table 2 indicate that pooling of the OR group and the TEVAR group is not feasible because of heterogeneity between groups.

• Among others, center should be one confounder. The number of patients per treatment group should be given.

• The main result needs to be derived from the multivariate analysis (L317ff).

• Limitation section: Please elaborate the effect of bias on the study results by given results from sensitivity analysis.

detailed comments:

L94: Here it si very important to make clear, whether enrollment of surgical and medicational patients is concurrent, same time period.

L140: Serious time trend bias is included by application of the treatments in two different time periods.

Here in particular similarity of follow-up times within groups should be given, (see L268)

L183, please avoid general statements and give the variables, which are analyzed by the stated method.

L273: Please use logrank test here, as this is a time to event study.

Please give statistical models which are used to derive the p-values in figure 2 and 3.

6. PLOS authors have the option to publish the peer review history of their article (what does this mean?). If published, this will include your full peer review and any attached files.

Reviewer #1: **Yes: **Rafik Margaryan

Reviewer #2: **Yes: **Mohammed Quader

Reviewer #3: No

---

## [Author Response · Author response to Decision Letter 0]

23 Dec 2023

Reviewer #1: Congratulations to authors for extremely interesting topic choice and writing accuracy.

In order to get the most out of scientific questions one should design a prospective randomized trial, but in this case it is extremely complicated due to the type of disease.

Materials

Why do you consider the TEVAR as a surgical group?

⇒We appreciate your comment. After reconsideration, we have categorized the patients who were previously grouped into the surgical group into the TEVAR group and the open aortic repair group for comparison (Line 43-45, 106-109, Figure 1).

Results:

It seems that the surgical group has higher comorbidity?

Had bigger aorta

Had high risk status

Medical Group had higher mortality in the follow up?

⇒We thank you for your comment. We noted some complications after operation, even after TEVAR. Although there was a difference in aortic diameter, as shown in Table 2, the preoperative condition did not indicate that the surgical group was particularly at a high risk. On the contrary, in the medication group, only antihypertensive treatment was continued without correcting the site of indication for treatment, and we also observed aortic-cause mortality among the deaths during the follow-up. 

Topic is super specialistic. I think you should compare the TEVAR with medical therapy alone, because it seems that some of the patients in the medical group could have been addressed to the TEVAR. Try to add propensity score matching to two groups of interest and do subanalysis.

⇒We appreciate your comment. We have changed it to a comparison among the three groups. We also performed propensity score matching as you indicated and added a comparison of outcomes for two of each of the three groups (Line 119-121, 194-204, 330-339, Table 3, Figure 2).

Reviewer #2: The authors studied an important question on patients with progressive dilatation of Type B dissection or have reached an aortic size over 5.5 cm with our without any symptoms. They noticed that in patients with aortic diameter of over 5.5 cm there is benefit in intervention, rest of them did not benefit from intervention.

The paper is well written and made a sincere effort to address the question given the long duration of study and heterogeneity of population as well as use of TEVAR after 2016.

Few suggestions to authors to make the paper more easy for readers to follow.

1. Create a table that shows inclusion and exclusion criteria, make it simple

⇒Thank you for this comment. We have revised Figure 1 and added the exclusion criteria to it.

2. limit abbreviations to minimum- as a rule if you are not using the acronym for more than five times, spell it out. It is easier to stay focused on the message.

⇒We appreciate your suggestion. We have removed the “AKI” and “IQR” abbreviations and spelled them out as you suggested (Line 64, 68).

3. while the syntax and grammar are OK the flow of information is not. Please use a service to make the whole paper revised to improve readability.

⇒Thank you for your comment. As English is not the first language of the authors, we have once again used an English editing service (Enago; www.enagp.jp) to improve readability.

I thank the authors for undertaking this important study.

Reviewer #3: The authors present the results of a cohort study of 106 consecutive patients consisting of 57 surgical patients and 49 medicational treated patients with UTBAD with respect to all cause mortality. They concluded, that there was difference in mortality between the surgical and medicational treated patients.

General comment:

The hypothesis, primary endpoint variables, statistical analysis and sample size justification are not aligned.

• The statement in Line 108implies, that the most severe case (inoperable) and the less severe cases (refusal for invasive treatment) were pooled in the medicational treated group. This heterogeneity may cause bias and may be explored by Kaplan Meier Curves in subgroups.

We appreciate this note. The description contained an error. Cases that were inoperable were due to truly high-risk situations, and we excluded most severe cases. The outpatient physician’s decision that surgery was inappropriate was based on advanced age and frailty. This has been corrected in the text (Line 109-111).

• Although this is an exploratory study, sample size justification in terms of stating the desired precision of the (crude) estimates should be given,

Thank you for this suggestion. As our study was exploratory, it was difficult to calculate the detection power or to specify the desired precision for the crude estimates in advance. Nevertheless, we maintained a sample size that was consistent with that of previous similar studies. We have added this statement to the limitations (Line 90-91, 461-464).

• The major problem of the study design is the missing randomization. This result in an unfair comparison of the treatment effects. The authors used multivariate models but the elaboration of confounders is incomplete. So uncertainty of the results might need further exploration.

We appreciate this comment. We believe the reviewer’s point is entirely correct. We have reconsidered the confounding factors and performed a multivariate analysis as you indicated (Line 194-197, 211-213, Table 4).

• The analysis of the surgical group splits in TEVAR and OR technique. If this is of primary interest, the treatment factor should be split in 3 groups, medicational, TEVAR, OR. Then the statistical analysis of the time to event data should be made accordingly. Even here, without randomization a sound multivariate model building should be applied.

Thank you for this comment. We believe the reviewer’s point is entirely correct. As indicated, we have compared the results among the three groups and performed further multivariate analysis (Line 43-45, 106-109, Figure 1, Table 4).

• Concerning the multivariate model building, one first should identify promising confounders, e.g Cox models for each confounder variable (see table 1 and 2). If p≤.25 one look to potential interactions. Of primary interest are the confounder times treatment group interactions as Cox regression. From this it is clear whether to include some interaction terms. The remaining set of all promising confounders and interaction terms give the whole Cox regression model. The interaction part is missing.

We appreciate your guidance. As you suggested, we have created confounder variables and performed a multivariate analysis based on it. We have added this information to the “Statistical Analysis” section (Line 194-197, 211-213).

• Some results in table 2 indicate that pooling of the OR group and the TEVAR group is not feasible because of heterogeneity between groups.

We appreciate your comment, and we agree with your point. We have also added a three-group analysis after propensity score matching with the variables, as shown in Table 2 (Line 119-121, 194-204, 330-339, Table 3, Figure 2).

• Among others, center should be one confounder. The number of patients per treatment group should be given.

Thank you for this note. We agree with you. We have added the number of patients in each center to Table 1. We have also included the center as a confounder in the multivariate analysis and propensity score matching (Line 199-200, 212-213, Table 4).

• The main result needs to be derived from the multivariate analysis (L317ff).

We agree with this important suggestion. We have based the study conclusion on what was derived from the multivariate analysis (Line 468-471).

• Limitation section: Please elaborate the effect of bias on the study results by given results from sensitivity analysis.

We appreciate your comment. We have included a new sensitivity analysis using propensity score matching in the three groups. The additional analysis reaffirms that the treatment method does not significantly affect survival, which is consistent with the findings from the multivariate analysis (Line 454-458).

detailed comments:

L94: Here it si very important to make clear, whether enrollment of surgical and medicational patients is concurrent, same time period.

Thank you for this suggestion. We have added the statement “patient enrollment in the OR, TEVAR, and medication groups occurred concurrently.” (Line 151-153)

L140: Serious time trend bias is included by application of the treatments in two different time periods.

Here in particular similarity of follow-up times within groups should be given, (see L268) 

We appreciate this comment and have addressed this concern by adding information regarding follow-up times within two different periods. The “Discussion” section now acknowledges the presence of a serious time trend bias (Line 377-379, 414-415).

L183, please avoid general statements and give the variables, which are analyzed by the stated method.

We appreciate this note, and we agree. We have revised this section to be more factual (Line 192-197).

L273: Please use logrank test here, as this is a time to event study.

Thank you for pointing this out. We agree with you and have used the log-rank test (Line 300, 306, 319).

Please give statistical models which are used to derive the p-values in figure 2 and 3.

Thank you for the suggestion. We have compared the length of time from the onset of aortic events to mortality using a log-rank test (Line 205-207).

---

## [Decision Letter · Decision Letter 1]

28 Jan 2024

Surgical treatment in the chronic phase for uncomplicated Stanford type B aortic dissection

PONE-D-23-24756R1

Dear Dr. Matsushita

We’re pleased to inform you that your manuscript has been judged scientifically suitable for publication and will be formally accepted for publication once it meets all outstanding technical requirements.

Kind regards,

Alessandro Leone, MD

Academic Editor

PLOS ONE

Additional Editor Comments (optional):

**Comments to the Author**

1. If the authors have adequately addressed your comments raised in a previous round of review and you feel that this manuscript is now acceptable for publication, you may indicate that here to bypass the “Comments to the Author” section, enter your conflict of interest statement in the “Confidential to Editor” section, and submit your "Accept" recommendation.

Reviewer #1: All comments have been addressed

Reviewer #3: All comments have been addressed

2. Is the manuscript technically sound, and do the data support the conclusions?

Reviewer #1: Yes

Reviewer #3: Yes

3. Has the statistical analysis been performed appropriately and rigorously? 

Reviewer #1: Yes

Reviewer #3: Yes

4. Have the authors made all data underlying the findings in their manuscript fully available?

Reviewer #1: No

Reviewer #3: Yes

5. Is the manuscript presented in an intelligible fashion and written in standard English?

Reviewer #1: Yes

Reviewer #3: Yes

6. Review Comments to the Author

Reviewer #1: Thank you for your contribution, all the comments were addressed. There some minor english mistypings to correct. Congratulation for the topic and excellent results.

Reviewer #3: The authors adressed all my comments adequately. The paper appears as a sound scientific peace of work.

7. PLOS authors have the option to publish the peer review history of their article (what does this mean?). If published, this will include your full peer review and any attached files.

Reviewer #1: **Yes: **Rafik Margaryan

Reviewer #3: No

---

## [Editor Report · Acceptance letter]

14 Feb 2024

PONE-D-23-24756R1 

PLOS ONE

Dear Dr. Matsushita, 

I'm pleased to inform you that your manuscript has been deemed suitable for publication in PLOS ONE. Congratulations! Your manuscript is now being handed over to our production team.

Kind regards, 

on behalf of

Dr. Alessandro Leone 

Academic Editor

PLOS ONE